# Diaphragmatic Activity and Respiratory Function Following C3 or C6 Unilateral Spinal Cord Contusion in Mice

**DOI:** 10.3390/biology11040558

**Published:** 2022-04-06

**Authors:** Afaf Bajjig, Pauline Michel-Flutot, Tiffany Migevent, Florence Cayetanot, Laurence Bodineau, Stéphane Vinit, Isabelle Vivodtzev

**Affiliations:** 1Inserm, UMR_S1158 Neurophysiologie Respiratoire Expérimentale et Clinique, Sorbonne Université, 75013 Paris, France; afaf.bajjig@sorbonne-universite.fr (A.B.); tiffanymigevent77@gmail.com (T.M.); florence.cayetanot@sorbonne-universite.fr (F.C.); laurence.bodineau@sorbonne-universite.fr (L.B.); 2Inserm, END-ICAP, Université Paris-Saclay, UVSQ, 78000 Versailles, France; pauline.michel-flutot@uvsq.fr (P.M.-F.); stephane.vinit@uvsq.fr (S.V.)

**Keywords:** spinal cord injury, contusion model, respiratory function, diaphragmatic activity, phrenic motoneurons

## Abstract

**Simple Summary:**

Tetraplegia is one of the most devastating conditions that an individual can sustain and affects more than 2.5 million people worldwide. Tetraplegia not only affects mobility but also impacts spontaneous breathing such that survivors can be rendered ventilator dependent and have increased mortality. Although no treatment can restore respiratory function after tetraplegia, there is a need for more exploratory studies defining therapeutic approaches to ameliorate respiratory decline in tetraplegia. Here, we studied two models of tetraplegia mimicking human forms of injury, using spinal cord contusion in mice, either above the third cervical metameric segment (C3 model) or below the sixth cervical metameric segment (C6 model) innervation of the phrenic nerves. These nerves are responsible for contraction of the diaphragm, the main inspiratory muscle. Using measurements of spontaneous breathing and muscle activity of the diaphragm, we found reduced diaphragmatic activity in both models, but only the C3 model led to reduced spontaneous breathing similar to what is seen in humans with tetraplegia. Moreover, we found a decline in basal contractility of the diaphragm in the C3 model only. We conclude that the C3 model is an appropriate model to explore interventions aimed at restoring breathing following tetraplegia.

**Abstract:**

The majority of spinal cord injuries (SCIs) are cervical (cSCI), leading to a marked reduction in respiratory capacity. We aimed to investigate the effect of hemicontusion models of cSCI on both diaphragm activity and respiratory function to serve as preclinical models of cervical SCI. Since phrenic motoneuron pools are located at the C3–C5 spinal level, we investigated two models of preclinical cSCI mimicking human forms of injury, namely, one above (C3 hemicontusion—C3HC) and one below phrenic motoneuron pools (C6HC) in wild-type swiss OF-1 mice, and we compared their effects on respiratory function using whole-body plethysmography and on diaphragm activity using electromyography (EMG). At 7 days post-surgery, both C3HC and C6HC damaged spinal cord integrity above the lesion level, suggesting that C6HC potentially alters C5 motoneurons. Although both models led to decreased diaphragmatic EMG activity in the injured hemidiaphragm compared to the intact one (−46% and −26% in C3HC and C6HC, respectively, both *p* = 0.02), only C3HC led to a significant reduction in tidal volume and minute ventilation compared to sham surgery (−25% and −20% vs. baseline). Moreover, changes in EMG amplitude between respiratory bursts were observed post-C3HC, reflecting a change in phrenic motoneuronal excitability. Hence, C3HC and C6HC models induced alteration in respiratory function proportionally to injury level, and the C3HC model is a more appropriate model for interventional studies aiming to restore respiratory function in cSCI.

## 1. Introduction

Spinal cord injury (SCI) prevalence is estimated at 500,000 individuals worldwide each year [1]. Most cases of SCI happen at the cervical level [2,3], leading not only to locomotor impairment but also marked respiratory dysfunction. Indeed, cervical SCI at or above the level of phrenic motor neurons (C3 to C5) interrupts descending medullary spinal respiratory pathways and causes complete or partial diaphragm paralysis. As a result, forced vital capacity and total lung capacity are proportionally reduced with an increasing level and severity of injury [4], and only mechanical ventilation can ensure sufficient breathing in patients with the highest level of injury (C4 or above) [5]. Notably, almost all tetraplegic patients use mechanical ventilation (MV) acutely after injury, and many remain dependent on it. Moreover, even when patients are weaned from diurnal MV, most of them (60%) are still dependent on nocturnal ventilation support due to sleep-disordered breathing [6]. Furthermore, efficient diaphragmatic activity is needed to ensure coughing and sneezing abilities, both crucial for airway clearance in order to prevent respiratory infections [7]. Hence, respiratory dysfunction after SCI not only compromises comfort and life quality but also increases morbidity and mortality rates in patients living with such high injuries [8]. 

Currently, there is no efficient treatment that can be used to fully restore respiratory function in cervical SCI. Implanted phrenic stimulation (diaphragm pacing) allows some patients to be weaned from MV [9], but this procedure is invasive and only possible when phrenic nerve conduction is preserved, therefore allowing less than 10% of patients with SCI above C4 to be treated with diaphragm pacing [10]. Similarly, the restoration of diaphragm innervation by using the nerve transfer approach is limited by the need for the donor and their axons to have previously been innervated by the respiratory centers [11,12], making this approach complicated and limiting. Nevertheless, a growing number of interventional studies using pharmacological strategies, such as the modulation of noradrenergic, serotonergic or dopaminergic neurotransmission, or neuromechanical devices, such as electrical stimulation, robotic assistance or even the brain–computer interface, are suggesting that respiratory neuroplasticity may be enhanced to improve spontaneous ventilation after SCI [13,14]. Moreover, few non-invasive therapeutics such as intermittent hypoxia protocols demonstrated their ability to induce substantial respiratory recovery in rodent preclinical models [15,16,17,18,19] of cervical injury as well as in human patients [20,21,22]. There is a need for more exploratory studies searching for a pharmacological agent, intervention or combinatorial approaches to further ameliorate the respiratory outcome following cervical SCI.

One of the most challenging aspects of medical research is ensuring that benefits found based on animal data can be translated to humans. Therefore, current and developing studies need representative preclinical models of cSCI to allow a better understanding of respiratory physiopathology after SCI and to unravel the benefit of therapeutic approaches. Currently, most studies investigating respiratory physiopathology, the neuroplasticity of medullary spinal axonal regeneration and phrenic motoneurons nuclei reinnervation following high SCI use the C2 hemisection in rat [22,23,24,25] and mouse [26,27,28] models. These studies have unraveled fundamental concepts of respiratory physiopathology after SCI [23,27] and provided bases for putative therapeutics to cure SCI [15,24,29]. However, section models are not ideal for investigating SCI pathophysiology because transections are scarce in clinical settings [24,29,30,31]. Contusion models may be more appropriate since they lead to spared motoneurons and not death of all neurons. Moreover, respiratory recovery after C2 injuries can be linked to cross-phrenic phenomena typical of C2 injury [32], whose existence is still debated in humans [33]. Hence, it may be relevant to use injury occurring at the phrenic motoneurons innervation level (C3–C5 in rodents) in a manner similar to that in previous publications [15,34,35,36,37,38,39,40]. Furthermore, despite recent progress in the generation of transgenic rats, there is currently almost no alternative to mice for studying the roles of specific genes after SCI. Indeed, when hypotheses are confirmed, further projects would need to explore and confirm the effect on genetically engineered lines. Hence, mouse models of cervical hemicontusion would be helpful to study the recovery of respiratory function after SCI with interventions.

To date, very few studies have sought to characterize the respiratory physiopathology in the mouse model of cervical hemicontusion. Of note, Charles Nicaise and colleagues previously reported diaphragm deficits and phrenic motor neuron degeneration following C4/C5 SCI in mice [41,42]. These were pioneering studies showing the diaphragmatic impact of cervical hemicontusion in mice. However, since phrenic nuclei are located in the C3–C5 region in mice [42,43,44], it is likely that some phrenic motoneurons at the C3 level survive after C4/5 injury. On the contrary, although complete T8 contusion decreases tidal volume and minute ventilation in rats [42], it is not known whether respiration dysfunction occurs after mid-cervical SCI without direct damage to the phrenic motoneuron (MN). Lastly, respiratory function has been measured following bilateral mid-cervical contusion injury at the C3/C4 level, but only in rats [39]. Therefore, it would be interesting to assess the effect of unilateral phrenic MN denervation after C3 hemicontusion as compared to intercostal and abdominal denervation after C6 hemicontusion in a mouse model of SCI. 

Furthermore, pulmonary volume has not been characterized after hemicervical contusion in mice. However, it is a direct, accurate and precise reflection of respiratory function and may be an interesting tool to investigate, especially given that it can be evaluated non-invasively at different time points and provide the functional effect of putative therapeutic strategies to restore impaired breathing. In the present study, we aimed to develop and investigate the effect of C3 and C6 hemicontusion models on both diaphragm activity and respiratory function to serve as preclinical models of cervical SCI in interventional studies with the aim of restoring respiratory function in SCI. 

## 2. Materials and Methods

### 2.1. Animals and Experimental Design

Adult WT Swiss OF-1 (Charles River, France) male mice of 20–30 g (10–12 weeks old) were housed in ventilated cages with controlled humidity, light and temperature. Animals were first housed in cages in groups of 5 before surgery and then in cages in groups of 2 after surgery. Only two animals needed to be separated due to showing aggressive behavior toward other males. *Ad libitum* food and water were provided in an enriched cage environment. All experiments reported in this manuscript conformed to the policies laid out by the Guide for the Care and Use of Laboratory Animals in the EU Directive 2010/63/EU for animal experiments. These experiments were approved by the Ethics Committee Charles Darwin CEEACD/N°5 (Project authorization APAFIS No. 2020070317412138 v3). 

Thirty-five animals were included in this study and were divided into 3 groups: (1) laminectomy (*n* = 5) or no surgery for one control mouse (sham/control, *n* = 6); (2) unilateral SCI at the C3 level (C3HC, *n* = 12); and (3) unilateral SCI at the C6 level (C6HC, *n* = 17). Two mice did not survive C3 hemicontusion and died during surgery, and one responded by self-mutilation on the 5th day and was excluded from the study. For all animals, terminal procedures were performed at J + 7 days post-lesion. From the 32 animals, 26 animals received plethysmography before and after surgery (*n* = 7 C3HC, *n* = 13 C6HC and *n* = 6 sham/control), and among these, 9 also received diaphragmatic electromyogram (EMG, *n* = 4 C3HC, *n* = 4 C6HC and *n*= 1 control) before being used for histologic analyses. In addition, 6 received EMG only (*n* = 2 C3HC and *n* = 4 C6HC) before being used for histologic analyses.

### 2.2. Unilateral Cervical Spinal Cord Contusion

Animals were placed in a closed chamber for anesthesia induction with isoflurane (4%) maintained throughout the procedure with a facial mask (1.5–2.5% isoflurane). As previously described in a study using a mouse model of cervical contusion targeting C4 and C5 levels [41], the dorsal skin and underlying muscles above the second cervical up to the first thoracic vertebrae were retracted. A dorsal laminectomy was performed to expose the spinal cord. A precision Impactor Device (RWD life science; 68,099 II) with a 1.5 mm tip impactor was used to perform the hemicontusion. The impactor parameters had a mean depth of 2.29 ± 0.41 mm, a speed of 2.41 ± 0.29 m/s and a dwell time of 0.50 ± 0.01 s (mean ± SEM) for the two injury types (not different for C3 and C6 epicenters). After contusion, sutures were used to close the wounds and skin. The isoflurane vaporizer was turned off, and the mice received subcutaneous injections of analgesia (buprenorphine, 0.1 mg/kg: QN02AE01, CEVA). After surgery, the animals were placed on a heated pad to recover. The animals were then placed in a cage containing water and a recovery diet gel, both in a small Petri dish, and then solid food was provided in the cage. Food and water were also provided on the top grid of the cage. Analgesia was maintained for at least 3 days following injury.

### 2.3. Breathing Recording Using Plethysmography

Before and 7 days after spinal cord contusion, respiratory variables were measured (tidal volume—V_T_, breathing frequency—*B**_f_*, minute ventilation—V_E_, inspiratory time—Ti and expiratory time—Te). One-hour plethysmography recordings were obtained for each mouse. The animals were placed in 400 mL chambers, breathing spontaneously and not constrained. Plethysmography does not require anesthesia and is a non-invasive procedure. The air in the chambers was constantly renewed by a pump at a flow rate of 0.5 L/min. The plethysmography chamber was hermetic, sealed and composed of 2 sections: a 150 mL reference chamber and a 250 mL chamber wherein the animal was placed. Before baseline measurements, the animals were placed in ventilated and enriched cages in the animal facility for one week for acclimatization. Subsequently, three-day plethysmography acclimation was conducted to familiarize the animals with the plethysmography chambers (Emka, France). This phase limited stress and resulted in calm animals, thereby allowing successful recordings with plethysmography software (Emka IOX software, Paris, France) on the 4th and 5th days of the week. The success rate (Sr) was obtained from the ratio of validated cycles (i.e., signals matching respiratory cycles) and the number of detected cycles for each 20 s period. Sr = 100% indicates that all detected respiratory cycles were validated by the software. We considered only measurements of respiratory function with Sr > 60. Mean Sr was similar between groups (76 ± 4, 75 ± 6 and 78 ± 10 in C3HC, C6HC and sham groups, respectively). Baseline recordings were performed for two consecutive days to check for data reproducibility. Post-surgery measurement was performed on day 7 to rule out any putative effect of analgesia. 

### 2.4. Electrophysiological Recording of the Diaphragm

Electromyogram (EMG) was performed on 7 days post-injury. Anesthesia was induced using intraperitoneal injection of ketamine (QN01AX03, Virbac, Nice, France) (100 mg/kg) and xylazine (QN05CM92, Bayer, Leverkusen, Germany) (10 mg/kg). The animals were placed supine on a heating pad to maintain physiologic and constant body temperature (37.5 ± 1 °C). A laparotomy was performed to expose the diaphragm. Electromyogram (EMG) of the crural portion of the diaphragm ipsilateral and contralateral to the spinal cord lesion was obtained via handmade bipolar surface silver electrodes placed on the muscles during spontaneous poïkilocapnic normoxic breathing. EMG was amplified (Model 1800; gain, 100; A-M Systems, Everett, WA, USA) and bandpass filtered (100 Hz–10 kHz). Signals were digitized with Powerlab data acquisition hardware/software (Acquisition rate: 4 k/s; AD Instruments, Dunedin, New Zealand) connected to a computer and analyzed using LabChart 7 Pro software (AD Instruments, Dunedin, New Zealand). EMG was integrated (50 ms decay). After the experiment, the animals were euthanized with exsanguination, followed by intracardiac perfusion of saline and 4% paraformaldehyde (4 °C) for tissue fixation and subsequent harvesting.

### 2.5. Tissue Processing

After fixation, the C1-C8 segment of the spinal cord was dissected and immediately placed in cold 4% paraformaldehyde (P6148 Sigma-Aldrich, Darmstadt, Germany) for 48 h and then cryoprotected in 30% sucrose (in 0.9% NaCl, S9888, Sigma-Aldrich, Darmstadt, Germany) for 48 h and stored at −80 °C. Frozen transversal (C1–C5 or C4–C8 spinal cord) free-floating sections (30 μm) were obtained using a Thermo Fisher CryoStar NX70 cryostat. Spinal cord sections were stored in a cryoprotectant solution (sucrose 30% (pharma grade, 141621, AppliChem, Darmstadt, Germany), ethylene glycol 30% (BP230-4, Fisher Scientific, Illkirch, France) and polyvinylpyrrolidone 40 (PVP40-100G; Sigma-Aldrich 1% in phosphate-buffered saline (PBS) 1X) (BP665-1; Fisher Scientific, Illkirch, France) at −22 °C. Every fifth section from C1–C5 or C4–C8 was used for lesion evaluation to examine the extent of injury using cresyl violet histochemistry: 10 min in cresyl violet solution (0.001% cresyl violet acetate (C5042-10G, Sigma-Aldrich, Darmstadt, Germany) and 0.125% glacial acetic acid (A/0400/PB15, Fisher Scientific, Illkirch, France) in distilled water), 1 min in 70% ethanol, 1 min in 95% ethanol, 2 × 1 min in 100% ethanol (E/0600DF/17, Fisher Scientific, Illkirch, France) and 2 min in xylene (X/0100/PB17, Fischer Scientific, Illkirch, France). Then, they were coverslipped using Eukitt^®^ mounting medium, and slide microphotographs were taken with a slide scanner (Aperio AT2, Leica, France).

### 2.6. Immunohistofluorescence

Immunohistodetection of markers of interest was performed on spinal cord samples from C3HC and C6HC for qualitative assessments. Free-floating transverse sections of the C1–C5 or C4–C8 spinal cord stored in cryoprotectant solution were washed 3 times in PBS 1X and placed in blocking solution (normal donkey serum (NDS) 5% and 0.2% Triton 100X in PBS 1X) for 30 min and then incubated with primary antibodies using ionized calcium-binding adaptor protein-1 (Iba1) (Abcam ab5076, 1/400, goat polyclonal) and glial fibrillary acidic protein (GFAP (Millipore-Merck AB5804, 1/4000, rabbit polyclonal) in blocking solution overnight in an orbital shaker at 4 °C. After 3 PBS 1X washes, sections were incubated in the corresponding secondary antibodies Alexa Fluor 488 Donkey anti-rabbit (Molecular Probes A21206, 1/2000, ThermoFisher, Rockford, IL, USA) and Alexa Fluor 647 Donkey anti-goat (Molecular Probes A21447, 1/2000, ThermoFisher, Rockford, IL, USA) for 2 h at room temperature, and they were then washed again 3 times with PBS 1X. Sections were then incubated with NeuroTrace™ 530/615 (N21482, Invitrogen, Waltham, MA, USA), a neuronal marker (Nissl stain), for 10 min and then washed again 3 times with PBS 1X. Images of the different sections were captured with a Hamamatsu ORCA-R^2^ camera mounted on an Olympus IX83 P2ZF scanning microscope.

### 2.7. Data Processing and Statistical Analyses

*EMG analyses*. The amplitude of at least 10 double-integrated diaphragmatic EMG inspiratory bursts during normoxia was calculated for each animal from the injured and the intact sides using LabChart 7 Pro software (AD Instruments). For inter-inspiratory burst basal signal analysis during normoxia, the amplitude of a minimum of 5 basal diaphragmatic EMG signals was calculated using LabChart 7.

*Statistical analyses*. Data are presented as mean ± standard deviation (SD) or median (interquartiles), and differences were considered significant at *p* < 0.05. GraphPad Prism software 9.3.1 was used for all statistical analyses. Normality of the data was checked with the Shapiro–Wilk test. Within-group comparisons (weight loss, EMG activity and respiratory variables) were performed using paired *t*-test or Wilcoxon’s test (for non-parametric data and for groups with *n* < 7 animals), and between-group comparisons were performed using 2-way analysis of variance (diaphragmatic EMG and interburst) with the type of lesion and side (injured vs. intact) as factors. Between-group comparisons for changes in weight, V_T_, *B**_f_* and V_E_ after 7 days were performed using 2-way repeated measures analysis of variance (ANOVA) with post hoc Bonferroni. The relationship between changes in V_T_ (post/pre ratio) and in hemidiaphragmatic activity (injured/intact EMG ratio) was assessed using simple regression analysis and Pearson coefficient correlation.

## 3. Results

### 3.1. Physiologic Effect of Cervical Contusion

Body weight loss was observed in both groups (after C3HC and C6HC), with no difference between groups in weight change at 7 days post-lesion (Table 1) being noted. After slight, daily decreases, body weight stabilized after the fifth day post-lesion in both groups. After C3HC, complete ipsilateral forelimb paralysis was observed, whereas partial paralysis was observed after C6HC.

### 3.2. Histological Analysis of C3 and C6 Hemicontusions

Histological analyses are shown in Figure 1 and Figure 2. Seven days after hemicontusion, the spinal cord transverse sections stained with cresyl violet (Nissl body labeling) displayed tissue damage at the site of injury following C3HC or C6HC (Figure 1A,B and Figure 2). GFAP-positive cells (astrocytes), Iba1-positive cells (microglia and macrophages) and NeuroTrace labeling (neurons) showed the anatomical cell composition of the site of injury in both models (Figure 1C,D). Note the absence of GFAP labeling at the site of injury, as well as the blurred NeuroTrace labeling and ameboid Iba1-positive cells at the site of injury. In addition, as shown in Figure 2, the extent of the lesion reached 600 µm rostral and caudal to the lesion epicenter for both C3HC and C6HC. 

### 3.3. Respiratory Function Following C3 or C6 Hemicontusion

Plethysmography measurements are presented in Figure 3 and Table 1. There was a significant reduction in V_T_ at 7 days post-surgery after C3HC when corrected for weight loss (post vs. pre: 10.1 ± 2.1 vs. 14.3 ± 3.6 µL/g, −25%) compared to C6HC (14.2 ± 2.50 vs. 14.9 ± 2.6 µL/g, −5%) and sham (17.9 ± 4.6 vs. 14.9 ± 2.6 µL/g, +15%) (RM ANOVA, *p* = 0.01, Figure 3). There was also an increase in *B_f_* in the C3HC group 7 days after surgery (273 ± 36 vs. 233 ± 25, *p* = 0.04), but differences were not observed in the two other groups. Lastly, changes in V_E_ were significantly different between the groups: 7 days after surgery, V_E_ was lower in C3HC than in sham but not C6HC (post vs. pre: 98 ± 22 vs. 139 ± 49 mL/min, −20%, *p* = 0.02) (Figure 3 and Table 1). 

### 3.4. Diaphragmatic Activity Following C3 or C6 Hemicontusion

Diaphragm activity is shown in Figure 4. Seven days post-lesion, the activity in the ipsilateral (injured) hemidiaphragm showed a greater reduction than that in the contralateral (intact) hemidiaphragm after both C3HC (0.78 ± 0.43 vs. 1.62 ± 0.68 A.U. normalized to control, −46% *p* = 0.02) and C6HC (0.88 ± 0.42 vs. 1.21 ± 0.53 A.U. normalized to sham, −*p* = 0.02). The reduction in the injured hemidiaphragm compared to the intact one after C3HC tended to be significantly more pronounced than after C6HC (−46% vs. −27%, 2 ways RM ANOVA, *p* = 0.07) and was associated with increased activity of the intact hemidiaphragm (Figure 4C). In addition, the EMG amplitude recordings of two inspiratory interbursts on the injured side of the C3 mice was increased at 7 days post-injury compared to the intact side and to the C6 recordings (Figure 4D; injured side: 18 ± 15 vs. intact side: 3 ± 4 mV in C3HC compared to injured side: 4 ± 5 vs. intact side: 4 ± 3 mV in C6HC; two-way ANOVA; *p* = 0.01). Lastly, we found a significant correlation between changes in tidal volume (hemicontusion group vs. intact group) and changes in diaphragm activity (ipsilateral hemidiaphragm vs. contralateral hemidiaphragm) (r = 0.72, *p* = 0.01, Figure 5).

## 4. Discussion

The present study compared for the first time two distinct models of cervical spinal cord injury differentially impacting the phrenic MN pool, and it demonstrates their respective effects on diaphragmatic activity and respiratory function in adult mice. Although the mice tolerated and survived after surgery similarly in both models, C3 hemicontusion had a more deleterious impact on both diaphragmatic activity and respiratory function, which were closely correlated. Moreover, we found a significant increase in interburst diaphragmatic activity in the C3HC model, suggesting impaired contractility. These rodent C3HC and C6HC injury models induce alterations in respiratory function proportionally to the level of injury, suggesting that the C3HC model is more appropriate for interventional studies aiming to restore respiratory function in cSCI.

Mid-cervical contusion models in mice were developed ten years ago by Nicaise and collaborators [41,42] but have been poorly studied since then due to the small animal size and challenging surgery procedures. Unilateral models allow for better survival of animals following surgery and reduce mortality rates. In the present study, we were interested in comparing two models of mid-cervical hemicontusion, which differ in terms of their impact on phrenic MNs. Indeed, the C3HC model may, in theory, lead to ipsilateral denervation below C3 and, hence, to the removal of the whole pool of phrenic MNs, reducing diaphragmatic activity, while C6HC would allow for preservation of the phrenic MN pool but may still impact the extra-diaphragmatic muscles involved in respiration. Our qualitative assessment of NeuroTrace on the lesion site 7 days after C3HC and C6HC (Figure 1C,D) showed unilateral destruction of neuron cell bodies and white matter in all animals, suggesting that neuronal conduction below the lesion site may be compromised in these animals. Moreover, GFAP and Iba1 staining was also limited to the border of the lesion site, suggesting that the injured tissue at 7 days post-HC was scar fibrotic tissue without any visible cystic cavities, as opposed to what has previously been observed in section or contusion in rat models [45,46,47]. Indeed, a recent study from our group reported increased expressions of Iba1 and GFAP after C2 hemisection in rats, suggesting that, similar to hemisection, hemicontusions may also induce recruitment and activation of microglia, macrophages and astrocytes 7 days post-HC around the lesion site [48]. 

Furthermore, as shown in Figure 2, both contusion types extended rostro-caudally. As a result, it is likely that not only does C6HC impact extra-diaphragmatic respiratory muscles via denervation of the conduction below the site of the lesion (intercostal denervation, for example), but it may also partly damage phrenic conduction at the C5 MN pool. Similar extension of the lesion (up to 600 µm) has recently been reported in mice after C5 hemicontusion [49], although other authors showed that double hemicontusion at C4 and C5 impacted their respective spinal segments [41,42]. Differences between studies may derive from technical aspects of the impactor device. We used a new pneumatic and electric system controlling for depth, velocity and dwell time of the impact. Nevertheless, despite a slight impact on the C5 spinal segment, C6HC caused considerably less damage to respiratory function than did C3HC. 

A primary finding of the present study is the reduction in tidal volume after C3HC compared to the sham lesion and to C6HC. Our results corroborate previous findings in rats. Choi et al. previously reported a reduction in V_T_ and V_E_ after C4/5 double hemicontusion in rats even when considering weight reduction after SCI [37]. Moreover, Warren et al. previously showed that spinal cord contusion at the C3 level leads to decreased V_T_ under “normal” breathing conditions at an acute stage, which is similar to the results of our study (7 days post-surgery), but also at chronic stages up to 3 weeks post-trauma [25]. Recently, Chiu et al. confirmed that mid-cervical contusion causes a significant reduction in V_T_ that remains at the chronic injury stage [50]. However, to the best of our knowledge our study shows for the first time the effect of mid-cervical contusion in a mouse model of C3 injury. Furthermore, since reduced ventilation was not found after C6HC, i.e., when phrenic MNs are less damaged (C5 spinal level), the present study strongly suggests that damaged phrenic MNs are responsible for the reduction in V_T_ volume during resting breathing in mice after hemicontusion. Moreover, *B_f_* increased after C3HC. This may be explained by an attempt to compensate for the reduced V_T_ to maintain V_E_, as demonstrated in other studies. Indeed, a similar change in the respiratory pattern (lower V_T_ and increased *B_f_*) has been reported in rats after C4/5 [37,38]. In our study, this increase was not sufficient to restore normal ventilation, which was still 20% lower than baseline values (Figure 3). Of note, C3HC led to a significant reduction in tidal volume compared to C6HC. This result suggests that the C3HC model has a more deleterious effect on respiratory function and would, therefore, be a better model to study the effect of SCI on ventilatory recovery with interventions.

In parallel with reduced respiratory function, C3HC was also associated with altered diaphragmatic activity. Indeed, we found a greater deficit in diaphragmatic EMG with a higher injury level, suggesting that less spontaneous recovery occurs after injury. This may be due to the fact that C3HC induced deafferentation of all the phrenic MNs on the injury side, leading to greater loss of phrenic MN drive and less diaphragmatic contraction on the injured side. Similarly, C3/C4 contusion in the rat induced permanent hemidiaphragm paralysis without inducing phrenic MN neuronal loss [51]. Moreover, a previous study reported permanent diaphragm deficit after C4/5 hemicontusion in mice and showed that it may last up to 6 weeks post-injury [41]. In addition, these results are similar to those obtained in rat models of C2 hemisection [24,52], suggesting that the C3HC model has effects comparable to those of the complete unilateral denervation of phrenic MN pools, allowing for investigation of respiratory function recovery without the potential interference of the crossed phrenic phenomenon [23,27]. In fact, diaphragm activity and pulmonary volume were closely related (Figure 5), such that the reduction in V_T_ after C3HC was proportional to the reduced activity of the injured hemidiaphragm. In other words, V_T_ measurement could be a good functional marker of reduced diaphragmatic activity 7 days post-injury in these mice.

Lastly, the increased background noise observed between bursts after C3HC may reflect a change in phrenic motoneuronal excitability that was not observed after C6HC. This phenomenon has been previously reported and attributed to phrenic MN desynchronization or even spastic activity on the injured side following cervical C2 hemisection in mice [26]. Indeed, C3HC interrupts inhibitory projections from the Bötzinger complex of expiratory neurons [53], which could increase MN excitability, leading to spontaneous diaphragmatic EMG firing during the expiratory phase. In the present study, the massive injury induced by the contusion at C3 could have led to a larger inflammatory response than that of the cervical C2 hemisection, leading to persistent interburst activity compared to the hemisected animals. In fact, this interburst activity disappeared 7 d post-hemisection in mice [26]. Longer C3 post-contusion time is needed to confirm this phenomenon and to better describe the pathophysiological mechanisms. 

## 5. Conclusions

The present study describes for the first time the comparative impact of C3 and C6 hemicontusions on both diaphragmatic activity and respiratory function. Our results suggest that the C3HC model has a more deleterious impact on spontaneous ventilation than the C6HC model, possibly due to greater damage to phrenic MN pools and diaphragmatic activity. Further studies are needed to confirm the motoneuronal loss and better explain the alteration in diaphragmatic activity; however, C3HC seems to be an appropriate model for interventional studies aiming to restore respiratory function in cSCI. Such models are critical to investigate respiratory neuroplasticity induced by pharmacological and/or interventional therapeutics, such as changes in neuronal networks, motoneuron excitability and synaptic connectivity. From the clinical perspective, this model will help in the identification of new treatment modalities to enable the restoration of respiratory function in cervical SCI.

## Figures and Tables

**Figure 1 biology-11-00558-f001:**
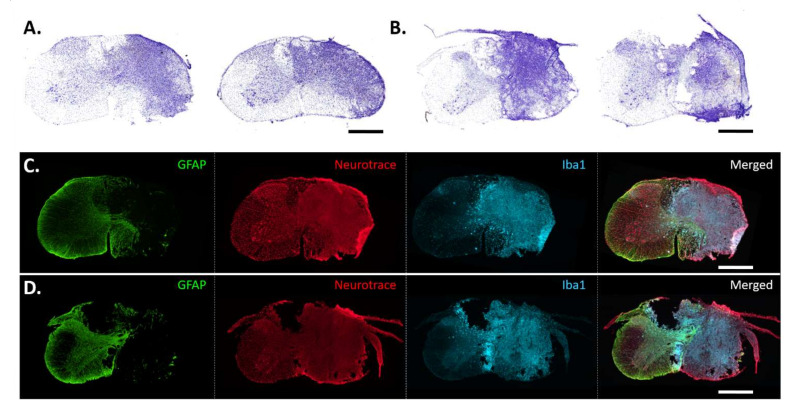
**Examples of extent of injury to C3 and C6 spinal segments in mice 7 days post-contusion**. Representative examples of cresyl violet staining at the injured site 7 days post-contusion for C3- (*n* = 6) (**A**) and C6- (*n* = 6) (**B**) injured groups. Representative images showing immunolabeling of glial fibrillary acidic protein (GFAP) (glial cells, in green), NeuroTrace (neurons in red) and ionized calcium-binding adaptor protein-1 (Iba1) (microglia, aqua) for C3-contused group (**C**) and C6-contused group (**D**). Note that half of the spinal cord is completely injured by the contusion for both groups, and no neurons remained in the scar. Scale bar: 500 µm.

**Figure 2 biology-11-00558-f002:**
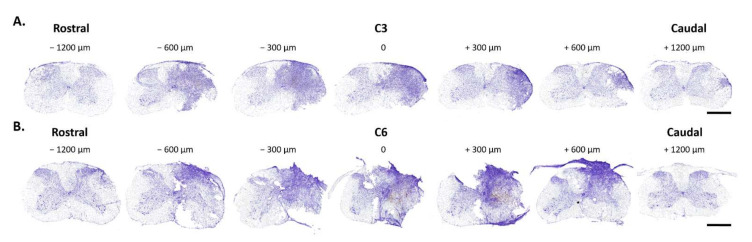
**Representative rostro-caudal extent of contusion injury in C3- (A) and C6- (B) contused groups**, where 0 represents the epicenter of the injury. Each picture is separated by 300 µm width. Note that the injury extends more than 2000 µm from the epicenter. Scale bar: 500 µm.

**Figure 3 biology-11-00558-f003:**
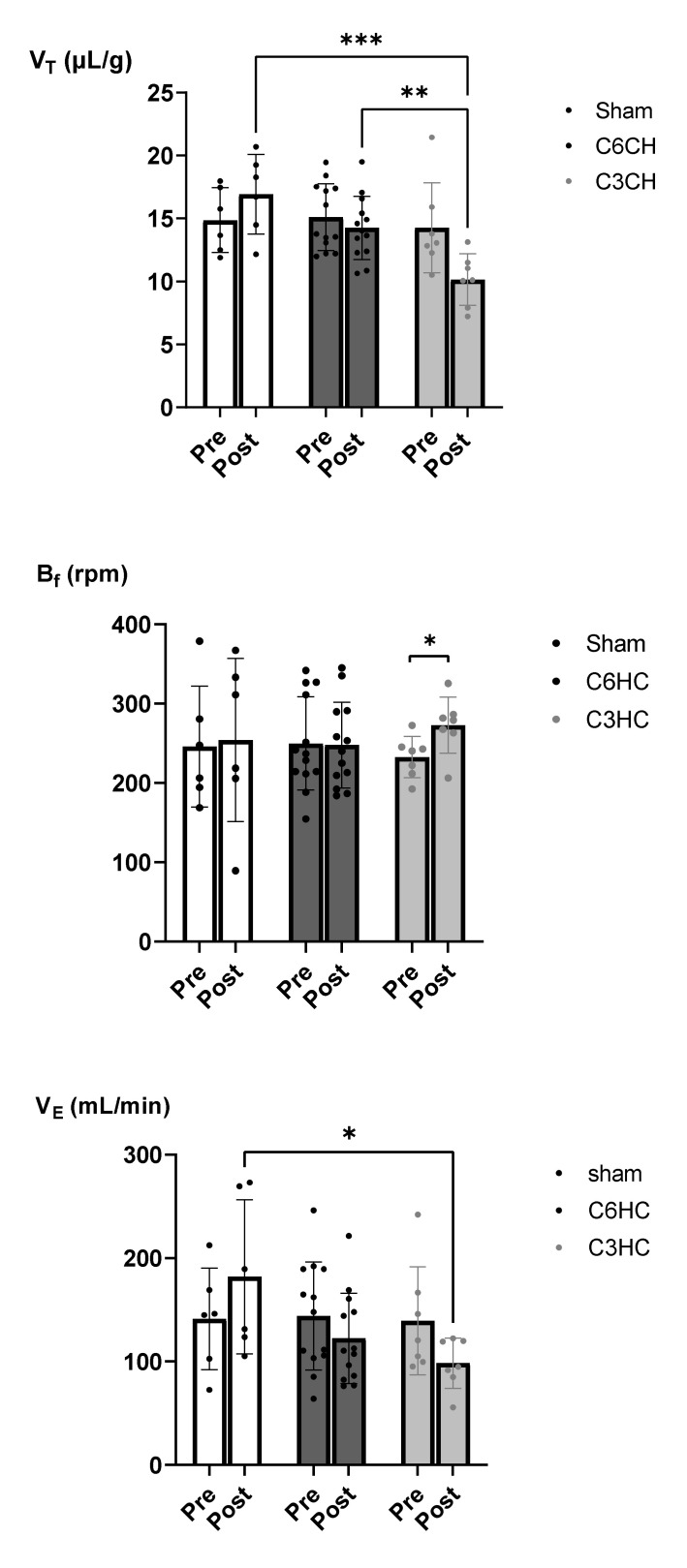
**Respiratory pattern following C3 and C6 hemicontusions (HCs).** Mean ± SD of tidal volume (V_T_), breathing frequency (*B_f_*) and minute ventilation (V_E_) before and 7 days after sham surgery, C6HC or C3HC. *** *p* < 0.001, ** *p* < 0.01, * *p* < 0.05. Bonferroni post hoc analysis: V_T_ at 7 days post-C3HC was significantly lower than in sham and C6CH groups, and VE at 7 days post-C3HC was significantly lower than in sham.

**Figure 4 biology-11-00558-f004:**
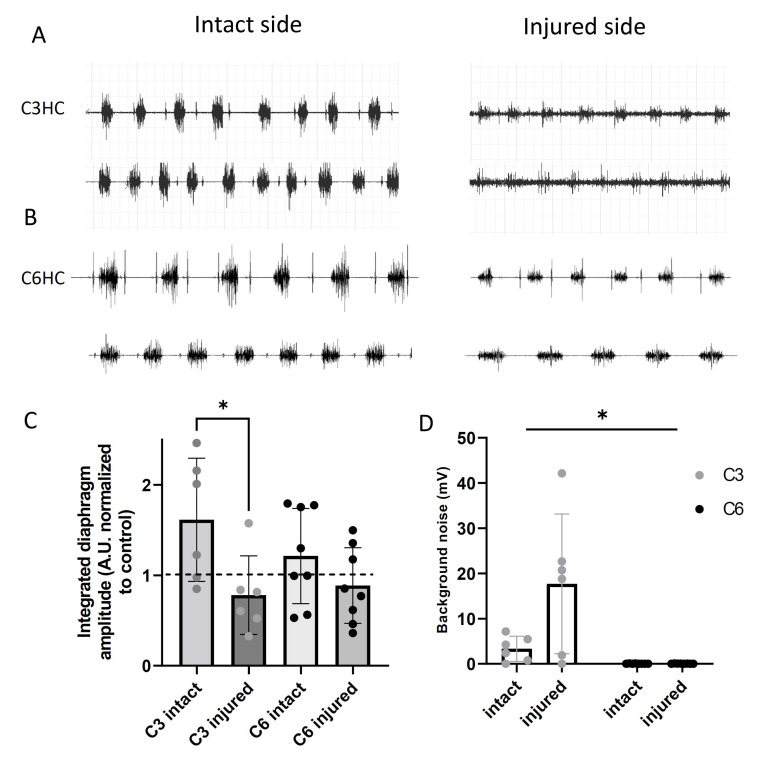
**Diaphragm activity following C3 and C6****hemicontusions (HCs).** Representative examples of diaphragmatic EMG in two mice 7 days after C3 hemicontusion (**A**) or C6 hemicontusion (**B**) and mean +/− SD in each group of animals (**C**). Amplitude recordings of 2 inspiratory interbursts and of the ECGs in injured and intact sides after C3 and C6 hemicontusions (**D**) * *p* < 0.05.

**Figure 5 biology-11-00558-f005:**
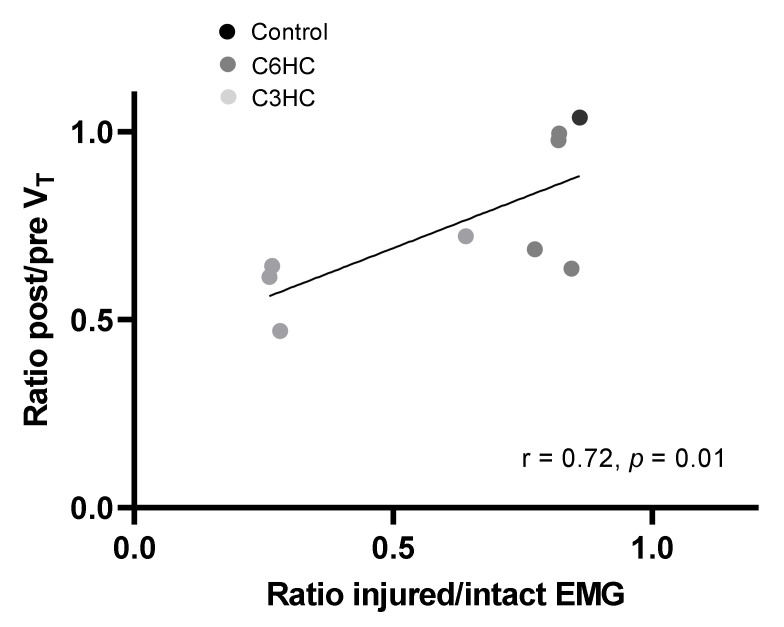
**Relationship between changes in respiratory function and diaphragmatic activity.** Simple linear relationship between diaphragmatic EMG activity of the injured side and that of the intact side (injured-to-intact ratio) and changes in tidal volume (ratio of post/pre values) 7 days post-lesion in animals after performing both plethysmography and EMG recordings (Pearson correlation coefficient).

**Table 1 biology-11-00558-t001:** Respiratory variables following C3 or C6 hemicontusion.

Variables		Sham	C3HC	C6HC
Weight (g)	day of surgery	38 (9)	41 ± 5	37 ± 3
	day 1	-	38 ± 3	35 ± 2
	day 2	-	36 ± 4	34 ± 2
	day 5	-	33 ± 3	33 ± 3
	day 7	39 (12)	34 ± 3	34 ± 3
V_T_ (µL/g)	pre	14.7 (5.2)	14.3 ± 3.6	14.9 ± 2.6
	7 d post	17.5 (5.7)	10.1 ^#ab^ ± 2.1	14.2 ± 2.5
V_T_ (mL)	pre	0.56 (0.11)	0.59 ± 0.2	0.56 ± 0.1
	7 d post	0.62 (0.20)	0.36 * ± 0.1	0.48 * ± 0.1
*B**_f_* (rpm)	pre	226 (117)	233 ± 25	250 ± 59
	7 d post	264 (165)	273 * ± 36	248 ± 54
V_E_ (mL/min)	pre	146 (85)	139 ± 52	144 ± 52
	7 d post	160 (151)	98 ^#a^ ± 24	122 ± 44
Ti (ms)	pre	107 (55)	111 ± 12	98 ± 21
	7 d post	98 * (58)	96 ± 14	90 ± 18
Te (ms)	pre	168 (82)	179 ± 26	171 ± 47
	7 d post	155 (90)	150 * ± 38	179 ± 47

Definitions of abbreviations: V_T_, tidal volume; *B_f_*, breathing frequency; V_E_, minute ventilation; Ti, inspiratory time; Te, expiratory time. * within-group comparison: *p* < 0.05. ^#^ between-group comparison *p* < 0.05. ^a^ post hoc comparison with sham: *p* < 0.01; ^b^ post hoc comparison with C6HC: *p* < 0.001.

## Data Availability

Data supporting reported results are available on request from the corresponding author.

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
