# Peer review of "Diaphragmatic Activity and Respiratory Function Following C3 or C6 Unilateral Spinal Cord Contusion in Mice"

_biology, 2022, doi:10.3390/biology11040558_

Round 1

Reviewer 1 Report

This work studied the diaphragmatic activity and respiratory function following C3 or C6 unilateral spinal cord contusion in mice. There are some concerns in this manuscript as follows:

  • Title: The word “function” should be replaced with “functions”
  • Abstract:

- The meaning of the abbreviations should be clearly defined at their first mention (e.g. WT).

- The objective of the study should be mentioned clearly in the abstract.

  • Introduction:

The novel points in this study should be clarified because there are previous studies that evaluated the respiratory functions following bilateral mid-cervical contusion injury. e.g. https://pubmed.ncbi.nlm.nih.gov/21963673/

  • Materials and methods:
  1. The exact source and CAT numbers of the used kits and chemicals should be mentioned.
  2. How did you know that the animals were acclimatized?
  3. The housing conditions of the animals should be mentioned in a more detailed manner.
  4. A reference for the method of “Unilateral cervical spinal cord contusion” should be added.
  5. In immunofluorescence, “Immunohistodetection of markers of interest were performed on 6 spinal cord samples from C3HC (n = 3) and C6HC (n = 3) for qualitative assessments”. Why wasn’t it performed in all animals to give more reliable data??
  6. In statistical analysis, what is meant by “SD”?
  • Results:
  1. Figures 1A, 1B and 2 are not clear. Please, replace them by more clear figures.
  2. The legends of the histopathological figures should be written in details to explain the microscopic changes that occur in the studied groups.
  • Discussion:

The discussion should not contain subheadings. Also, it should be summarized to focus on analysis of the results of the present study.

  • Conclusion:

- I think that the conclusion was not sufficient. The possible clinical implications of the results of the present study should be clearly addressed.

  • References: The format of the references should be unified.
  • General comments: The manuscript should be revised by English-naïve speaker to improve the quality of the language.

Author Response

Response to the reviewers

Biology #1649005

We thank the reviewers for their positive statement on our work and their valuable suggestions. The following is a point by point response to the comments along with the location of the resulting changes to the manuscript.

Reviewer 1:

This work studied the diaphragmatic activity and respiratory function following C3 or C6 unilateral spinal cord contusion in mice. There are some concerns in this manuscript as follows:

We thank the reviewer for his/her time and effort to review the paper and his/her fruitful suggestions.

C1: Title: The word “function” should be replaced with “functions”

R1: Here we are referring to the respiratory function so we do not believe that “function” should be replaced by “functions”.

C2: Abstract: The meaning of the abbreviations should be clearly defined at their first mention (e.g. WT). The objective of the study should be mentioned clearly in the abstract.

R2: Thank you for this suggestion. We defined first mentioned abbreviations and added the explicit objective: “We aimed to investigate the effect of hemi-contusion models of cSCI on both diaphragm activity and respiratory function to serve as pre-clinical models of cervical SCI”.

C3: Introduction: The novel points in this study should be clarified because there are previous studies that evaluated the respiratory functions following bilateral mid-cervical contusion injury. e.g. https://pubmed.ncbi.nlm.nih.gov/21963673/

R3: Thank you for this suggestion. We added the reference and we added a sentence in the introduction to mention this study and to highlight the novelty of the study (page 5).

Lastly, respiratory function had been measured following bilateral mid-cervical contusion injury at the C3/C4 level but only in rats [42]. Therefore, it would be interesting to assess the effect of unilateral phrenic MN denervation after C3 hemi-contusion as compared to intercostal and abdominal denervation after C6 hemi-contusion in a mouse model of SCI.”

C4: Materials and methods:

The exact source and CAT numbers of the used kits and chemicals should be mentioned.

R4.1: We added the CAT numbers and sources for all the kits and chemicals, as requested by the reviewer.

How did you know that the animals were acclimatized?

R4.2: Acclimation was checked using a success score (Sr), a number given by the plethysmography (IOX) software, reaches SR > 60.  “A Success rate (Sr) was obtained from the ratio of validated cycles (i.e. signals matching respiratory cycles -and not noise related to movement of the animal in the chamber for example-) and the number of detected cycles for each 20 second period.” We have added this information in the methods (page 8).

The housing conditions of the animals should be mentioned in a more detailed manner.

R4.3: We have added information related to housing page 6. Animals were first housed by cage of 5 before the surgery and then by cage of 2 after surgery. Only two animals needed to be separated due of aggressivity of the other male”

A reference for the method of “Unilateral cervical spinal cord contusion” should be added.

R4.4: The reference has been added in the method section page 6 (line 154).

R4.5: In immunofluorescence, “Immunohistodetection of markers of interest were performed on 6 spinal cord samples from C3HC (n = 3) and C6HC (n = 3) for qualitative assessments”. Why wasn’t it performed in all animals to give more reliable data??

R4.5: Because of the deadline of the special issue in Biology, at the time of submission, we did not perform the analysis for all animals but we have now performed analyses for all animals of each group found the same results. We deleted the “n=3” in the text and provided the number of animals in figure 1.

In statistical analysis, what is meant by “SD”?

R4.6: SD is for standard deviation, we have added the meaning of this abbreviation now

C5: Results:

C5.1: Figures 1A, 1B and 2 are not clear. Please, replace them by more clear figures.

We thank the reviewer for this comment. We made the figure clearer with a homogenous background.

C5.2: The legends of the histopathological figures should be written in details to explain the microscopic changes that occur in the studied groups.

We thank the reviewer for this comment. However, since we already described in details the microscopic changes in the results part main body, we think adding more info in the legend will be redundant with the result section.

C6: Discussion: The discussion should not contain subheadings. Also, it should be summarized to focus on analysis of the results of the present study.

R6: We thank the reviewer for this comment. We deleted the subheadings throughout the discussion part. However, since we wanted to put our results in the literature context, we chose to focus on what has been published by other laboratories rather than to solely focus on our own findings (which has been already discussed in the initial version of the manuscript).

C7: Conclusion: I think that the conclusion was not sufficient. The possible clinical implications of the results of the present study should be clearly addressed.

R7: Thank you for your suggestion. We have now added clinical implications in the conclusion. “Such models are critical to investigate respiratory neuroplasticity induced by pharmacological and/or interventional treatments, such as changes in neuronal networks, motoneurons excitability or synaptic connectivity, for instance. To a clinical perspective, it will help for identification of new treatments modalities enable to restore respiratory function in cervical SCI.”

C8: References: The format of the references should be unified.

R8: All references have been re-formatted using the biology End-note style.

C9: General comments: The manuscript should be revised by English-naïve speaker to improve the quality of the language.

R9: The entire manuscript had been edited by an English native speaker: Dr J Andrew Taylor (Associate professor, Harvard Medical School).

Reviewer 2 Report

Manuscript entitled „ Diaphragmatic activity and respiratory function following C3 or C6 unilateral spinal cord contusion in mice” is very interesting, well-written and well-planned experimental work. However, the text needs some corrections according to the following comments:

Introduction

line 49- In the text, reference numbers should be placed in square brackets [ ], use this rule for all citations in the manuscript

Materials and Methods

line 174 – explain an abbreviation IOX

line 179 – put dot instead comma

line 184, 188 – use only abbreviation EMG. It was explained in full name earlier in the text

line 218 – explain in full name the abbreviations Iba1 and GFAP

Results

line 271 – should be Bf instead of fB

line 275 – what if means in that sentence, shouldn't it be of?

Figure 3 - explain the full name of the abbreviations that are first used in the description of Figures or Tables

line 315, 316 – put dot after vs

Figure 4 – should be hemi-contusion in all parts of sentence

Discussion

341 – use only abbreviation MN

line 355 - put the publication on the list of References and assign it an appropriate number

line 371 – put dot after al

line 384 – should be VT

line 421 – write days instead d

line 423 – write physio-pathological

References

Please standardize the font and prepare References in accordance with the requirements specified by the journal as follows:

In the text, reference numbers should be placed in square brackets [ ] and placed before the punctuation; for example [1], [1–3] or [1,3]. For embedded citations in the text with pagination, use both parentheses and brackets to indicate the reference number and page numbers; for example [5] (p. 10). or [6] (pp. 101–105).

References should be described as follows, depending on the type of work:

Journal Articles:

  1. Author 1, A.B.; Author 2, C.D. Title of the article. Abbreviated Journal Name Year, Volume, page range.

DOI numbers (Digital Object Identifier) are not mandatory but highly encouraged.

Author Response

Response to the reviewers

Biology #1649005

We thank the reviewers for their positive statement on our work and their valuable suggestions. The following is a point by point response to the comments along with the location of the resulting changes to the manuscript.

Reviewer 2:

Manuscript entitled „ Diaphragmatic activity and respiratory function following C3 or C6 unilateral spinal cord contusion in mice” is very interesting, well-written and well-planned experimental work. However, the text needs some corrections according to the following comments:

We thank the reviewer for his/her time and effort to review the paper and his/her fruitful suggestions.

C1: Introduction

line 49- In the text, reference numbers should be placed in square brackets [ ], use this rule for all citations in the manuscript

R1: All references have been re-formatted using the biology End-note style.

C2: Materials and Methods

line 174 – explain an abbreviation IOX

line 179 – put dot instead comma

line 184, 188 – use only abbreviation EMG. It was explained in full name earlier in the text

line 218 – explain in full name the abbreviations Iba1 and GFAP

R2: Thank you for your suggestions. We have corrected all mistakes/omissions excepted for “line 179” (now 188) where we think that a comma is well needed (“76 ± 4, 75 ± 6 and 78 ± 10 in C3HC, C6HC and sham groups, respectively)”.

C3: Results

line 271 – should be Bf  instead of fB

line 275 – what if means in that sentence, shouldn't it be of?

Figure 3 - explain the full name of the abbreviations that are first used in the description of Figures or Tables

line 315, 316 – put dot after vs

Figure 4 – should be hemi-contusion in all parts of sentence

R3: Thank you for your suggestions. We made all changes.

C4: Discussion

341 – use only abbreviation MN

line 355 - put the publication on the list of References and assign it an appropriate number

line 371 – put dot after al

line 384 – should be VT

line 421 – write days instead d

line 423 – write physio-pathological

R4: Thank you for your suggestions. We made all changes.

C5: References

Please standardize the font and prepare References in accordance with the requirements specified by the journal as follows:

In the text, reference numbers should be placed in square brackets [ ] and placed before the punctuation; for example [1], [1–3] or [1,3]. For embedded citations in the text with pagination, use both parentheses and brackets to indicate the reference number and page numbers; for example [5] (p. 10). or [6] (pp. 101–105).

References should be described as follows, depending on the type of work:

Journal Articles:

  1. Author 1, A.B.; Author 2, C.D. Title of the article. Abbreviated Journal Name Year, Volume, page range.

DOI numbers (Digital Object Identifier) are not mandatory but highly encouraged.

R5: All references have been re-formatted using the biology End-note style.

Reviewer 3 Report

The comments attached

Author Response

Response to the reviewers

Biology #1649005

We thank the reviewers for their positive statement on our work and their valuable suggestions. The following is a point by point response to the comments along with the location of the resulting changes to the manuscript.

Reviewer 3:

Comments:
The spinal cord injury is the most notorious form of traumatic damage and nerve impairments which results in temporary or permanent loss of motor function, sensory function as well autonomic functions. Indeed, the authors have conducted the well-planned experimental study on
the Diaphragmatic activity and respiratory function following C3 or C6 unilateral spinal cord contusion in mice.

 We thank the reviewer for his/her time and effort to review the paper and his/her fruitful suggestions.

The authors need to address following concerns:

C1: In the experimental design, the authors have had three group of the mice control (sham), C3 and C6 and the number of animals were included in sufficiently manner, During the experimental protocol mortality of 2 mice were reported in C3 group, at what day of post-surgery contusion the animal were died and can the author provide the pathological reason of the mortality.

R1: Thank you for your comment. We have added this information in the manuscript page 6.

C2. In figures 1 and 2 the authors have provided the histological results and data analysis of C3 and
C6 groups, whereas as to proper systematic evaluation of this study the histological data of
control sham group animals need to be provide.

R2: We thank the reviewer for this suggestion. However, since the main goal of this manuscript is to compare 2 different contusion models, we did not investigate the sham group for Iba1 and GFAP immunostaining. We investigated the fact that our injuries will be similar between the 2 studied groups, and not the comparison between non-injured vs injured animals in term of “neuroinflammatory processes” (which will be a future study).

C3. The authors will how predict the chances of ARDS (acute respiratory distress syndrome) due to C3 and C6 spinal cord contusion consequences.

R3: We thank the reviewer for this comment. We did not observe ARDS in this study in the hours or days following contusions. This may be explained because the contusion is unilateral and hence allows for respiratory compensation with the intact side. Two animals died immediately during the surgery due to unexpected bilateral contusions rather than unilateral contusions.

C4: The compromised spinal cord contusion conditions would how impact the cardio-pulmonary pathophysiology.

R4: This is true that spinal cord contusion may compromise cardio-pulmonary physiology in the animals. We agree that this is a very interesting field of research but it was outside the scope of the present study which was focused on the respiratory neurophysiological impact of cSCI.